# Predictive State Recurrent Neural Networks

**Carlton Downey**
Carnegie Mellon University
Pittsburgh, PA 15213
cmdowney@cs.cmu.edu

**Ahmed Hefny**
Carnegie Mellon University
Pittsburgh, PA, 15213
ahefny@cs.cmu.edu

**Boyue Li**
Carnegie Mellon University
Pittsburgh, PA, 15213
boyue@cs.cmu.edu

**Byron Boots**
Georgia Tech
Atlanta, GA, 30332
bboots@cc.gatech.edu

**Geoff Gordon**
Carnegie Mellon University
Pittsburgh, PA, 15213
ggordon@cs.cmu.edu

## Abstract

We present a new model, Predictive State Recurrent Neural Networks (PSRNNs), for filtering and prediction in dynamical systems. PSRNNs draw on insights from both Recurrent Neural Networks (RNNs) and Predictive State Representations (PSRs), and inherit advantages from both types of models. Like many successful RNN architectures, PSRNNs use (potentially deeply composed) bilinear transfer functions to combine information from multiple sources. We show that such bilinear functions arise naturally from state updates in Bayes filters like PSRs, in which observations can be viewed as gating belief states. We also show that PSRNNs can be learned effectively by combining Backpropogation Through Time (BPTT) with an initialization derived from a statistically consistent learning algorithm for PSRs called two-stage regression (2SR). Finally, we show that PSRNNs can be factorized using tensor decomposition, reducing model size and suggesting interesting connections to existing multiplicative architectures such as LSTMs and GRUs. We apply PSRNNs to 4 datasets, and show that we outperform several popular alternative approaches to modeling dynamical systems in all cases.

## 1   Introduction

Learning to predict temporal sequences of observations is a fundamental challenge in a range of disciplines including machine learning, robotics, and natural language processing. While there are a wide variety of different approaches to modelling time series data, many of these approaches can be categorized as either recursive Bayes Filtering or Recurrent Neural Networks.

Bayes Filters (BFs) [1] focus on modeling and maintaining a belief state: a set of statistics, which, if known at time $t$, are sufficient to predict all future observations as accurately as if we know the full history. The belief state is generally interpreted as the statistics of a distribution over the latent state of a data generating process, conditioned on history. BFs recursively update the belief state by conditioning on new observations using Bayes rule. Examples of common BFs include sequential filtering in Hidden Markov Models (HMMs) [2] and Kalman Filters (KFs) [3].

Predictive State Representations [4] (PSRs) are a variation on Bayes filters that do not define system state explicitly, but proceed directly to a representation of state as the statistics of a distribution of features of *future observations*, conditioned on history. By defining the belief state in terms of observables rather than latent states, PSRs can be easier to learn than other filtering methods [5–7]. PSRs also support rich functional forms through kernel mean map embeddings [8], and a natural interpretation of model update behavior as a gating mechanism. This last property is not unique to

PSRs, as it is also possible to interpret the model updates of other BFs such as HMMs in terms of gating.

Due to their probabilistic grounding, BFs and PSRs possess a strong statistical theory leading to efficient learning algorithms. In particular, method-of-moments algorithms provide consistent parameter estimates for a range of BFs including PSRs [5, 7, 9–11]. Unfortunately, current versions of method of moments initialization restrict BFs to relatively simple functional forms such as linear-Gaussian (KFs) or linear-multinomial (HMMs).

Recurrent Neural Networks (RNNs) are an alternative to BFs that model sequential data via a parameterized internal state and update function. In contrast to BFs, RNNs are directly trained to minimize output prediction error, without adhering to any axiomatic probabilistic interpretation. Examples of popular RNN models include Long-Short Term Memory networks [12] (LSTMs), Gated Recurrent Units [13] (GRUs), and simple recurrent networks such as Elman networks [14].

RNNs have several advantages over BFs. Their flexible functional form supports large, rich models. And, RNNs can be paired with simple gradient-based training procedures that achieve state-of-the-art performance on many tasks [15]. RNNs also have drawbacks however: unlike BFs, RNNs lack an axiomatic probabilistic interpretation, and are therefore difficult to analyze. Furthermore, despite strong performance in some domains, RNNs are notoriously difficult to train; in particular it is difficult to find good initializations.

In summary, RNNs and BFs offer complementary advantages and disadvantages: RNNs offer rich functional forms at the cost of statistical insight, while BFs possess a sophisticated statistical theory but are restricted to simpler functional forms in order to maintain tractable training and inference. By drawing insights from both Bayes Filters *and* RNNs we develop a novel hybrid model, Predictive State Recurrent Neural Networks (PSRNNs). Like many successful RNN architectures, PSRNNs use (potentially deeply composed) bilinear transfer functions to combine information from multiple sources. We show that such bilinear functions arise naturally from state updates in Bayes filters like PSRs, in which observations can be viewed as gating belief states. We show that PSRNNs directly generalize discrete PSRs, and can be learned effectively by combining Backpropagation Through Time (BPTT) with an approximately consistent method-of-moments initialization based on two-stage regression. We also show that PSRNNs can be factorized using tensor decomposition, reducing model size and suggesting interesting connections to existing multiplicative architectures such as LSTMs.

## 2   Related Work

It is well known that a principled initialization can greatly increase the effectiveness of local search heuristics. For example, Boots [16] and Zhang et al. [17] use subspace ID to initialize EM for linear dyanmical systems, and Ko and Fox [18] use N4SID [19] to initialize GP-Bayes filters.

Pasa et al. [20] propose an HMM-based pre-training algorithm for RNNs by first training an HMM, then using this HMM to generate a new, simplified dataset, and, finally, initializing the RNN weights by training the RNN on this dataset.

Belanger and Kakade [21] propose a two-stage algorithm for learning a KF on text data. Their approach consists of a spectral initialization, followed by fine tuning via EM using the ASOS method of Martens [22]. They show that this approach has clear advantages over either spectral learning or BPTT in isolation. Despite these advantages, KFs make restrictive linear-Gaussian assumptions that preclude their use on many interesting problems.

Downey et al. [23] propose a two-stage algorithm for learning discrete PSRs, consisting of a spectral initialization followed by BPTT. While that work is similar in spirit to the current paper, it is still an attempt to optimize a BF using BPTT rather than an attempt to construct a true hybrid model. This results in several key differences: they focus on the discrete setting, and they optimize only a subset of the model parameters.

Haarnoja et al. [24] also recognize the complementary advantages of Bayes Filters and RNNs, and propose a new network architecture attempting to combine some of the advantages of both. Their approach differs substantially from ours as they propose a network consisting of a Bayes Filter concatenated with an RNN, which is then trained end-to-end via backprop. In contrast our entire network architecture has a dual interpretation as both a Bayes filter and a RNN. Because of this,

our entire network can be initialized via an approximately consistent method of moments algorithm, something not possible in [24].

Finally, Kossaifi et al. [25] also apply tensor decomposition in the neural network setting. They propose a novel neural network layer, based on low rank tensor factorization, which can directly process tensor input. This is in contrast to a standard approach where the data is flattened to a vector. While they also recognize the strength of the multilinear structure implied by tensor weights, both their setting and their approach differ from ours: they focus on factorizing tensor input data, while we focus on factorizing parameter tensors which arise naturally from a kernelized interpretation of Bayes rule.

## 3 Background

### 3.1 Predictive State Representations

Predictive state representations (PSRs) [4] are a class of models for filtering, prediction, and simulation of discrete time dynamical systems. PSRs provide a compact representation of a dynamical system by representing state as a set of predictions of features of future observations.

Let $f_t = f(o_{t:t+k-1})$ be a vector of features of future observations and let $h_t = h(o_{1:t-1})$ be a vector of features of historical observations. Then the predictive state is $q_t = q_{t|t-1} = E[f_t \mid o_{1:t-1}]$. The features are selected such that $q_t$ determines the distribution of future observations $P(o_{t:t+k-1} \mid o_{1:t-1})$.[1] Filtering is the process of mapping a predictive state $q_t$ to $q_{t+1}$ conditioned on $o_t$, while prediction maps a predictive state $q_t = q_{t|t-1}$ to $q_{t+j|t-1} = E[f_{t+j} \mid o_{1:t-1}]$ without intervening observations.

PSRs were originally developed for discrete data as a generalization of existing Bayes Filters such as HMMs [4]. However, by leveraging the recent concept of Hilbert Space embeddings of distributions [26], we can embed a PSR in a Hilbert Space, and thereby handle continuous observations [8]. Hilbert Space Embeddings of PSRs (HSE-PSRs) [8] represent the state as one or more nonparametric conditional embedding operators in a Reproducing Kernel Hilbert Space (RKHS) [27] and use Kernel Bayes Rule (KBR) [26] to estimate, predict, and update the state.

For a full treatment of HSE-PSRs see [8]. Let $k_f, k_h, k_o$ be translation invariant kernels [28] defined on $f_t$, $h_t$, and $o_t$ respectively. We use Random Fourier Features [28] (RFF) to define projections $\phi_t = RFF(f_t)$, $\eta_t = RFF(h_t)$, and $\omega_t = RFF(o_t)$ such that $k_f(f_i, f_j) \approx \phi_i^T \phi_j$, $k_h(h_i, h_j) \approx \eta_i^T \eta_j$, $k_o(o_i, o_j) \approx \omega_i^T \omega_j$. Using this notation, the HSE-PSR predictive state is $q_t = E[\phi_t \mid o_{t:t-1}]$. Formally an HSE-PSR (hereafter simply referred to as a PSR) consists of an initial state $b_1$, a 3-mode update tensor $W$, and a 3-mode normalization tensor $Z$. The PSR update equation is

$$q_{t+1} = (W \times_3 q_t)(Z \times_3 q_t)^{-1} \times_2 o_t. \tag{1}$$

where $\times_i$ is tensor multiplication along the $i$th mode of the preceding tensor. In some settings (such as with discrete data) it is possible to read off the observation probability directly from $W \times_3 q_t$; however, in order to generalize to continuous observations with RFF features we include $Z$ as a separate parameter.

### 3.2 Two-stage Regression

Hefny et al. [7] show that PSRs can be learned by solving a sequence of regression problems. This approach, referred to as *Two-Stage Regression* or 2SR, is fast, statistically consistent, and reduces to simple linear algebra operations. In 2SR the PSR model parameters $q_1$, $W$, and $Z$ are learned using

the history features $\eta_t$ defined earlier via the following set of equations:

$$q_1 = \frac{1}{T} \sum_{t=1}^{T} \phi_t \tag{2}$$

$$W = \left( \sum_{t=1}^{T} \phi_{t+1} \otimes \omega_t \otimes \eta_t \right) \times_3 \left( \sum_{t=1}^{T} \eta_t \otimes \phi_t \right)^{+} \tag{3}$$

$$Z = \left( \sum_{t=1}^{T} \omega_t \otimes \omega_t \otimes \eta_t \right) \times_3 \left( \sum_{t=1}^{T} \eta_t \otimes \phi_t \right)^{+}. \tag{4}$$

Where $+$ is the Moore-Penrose pseudo-inverse. It's possible to view (2–4) as first estimating predictive states by regression from history (stage 1) and then estimating parameters $W$ and $Z$ by regression among predictive states (stage 2), hence the name Two-Stage Regression; for details see [7]. Finally in practice we use ridge regression in order to improve model stability, and minimize the destabilizing effect of rare events while preserving consistency. We could instead use nonlinear predictors in stage 1, but with RFF features, linear regression has been sufficient for our purposes.[2] Once we learn model parameters, we can apply the filtering equation (1) to obtain predictive states $q_{1:T}$.

### 3.3   Tensor Decomposition

The tensor Canonical Polyadic decomposition (CP decomposition) [29] can be viewed as a generalization of the Singular Value Decomposition (SVD) to tensors. If $T \in \mathbb{R}^{(d_1 \times \ldots \times d_k)}$ is a tensor, then a CP decomposition of $T$ is:

$$T = \sum_{i=1}^{m} a_i^1 \otimes a_i^2 \otimes \ldots \otimes a_i^k$$

where $a_i^j \in \mathbb{R}^{d_j}$ and $\otimes$ is the Kronecker product. The rank of $T$ is the minimum $m$ such that the above equality holds. In other words, the CP decomposition represents $T$ as a sum of rank-1 tensors.

## 4   Predictive State Recurrent Neural Networks

In this section we introduce Predictive State Recurrent Neural Networks (PSRNNs), a new RNN architecture inspired by PSRs. PSRNNs allow for a principled initialization and refinement via BPTT. The key contributions which led to the development of PSRNNs are: 1) a new normalization scheme for PSRs which allows for effective refinement via BPTT; 2) the extention of the 2SR algorithm to a multilayered architecture; and 3) the optional use of a tensor decomposition to obtain a more scalable model.

### 4.1   Architecture

The basic building block of a PSRNN is a 3-mode tensor, which can be used to compute a bilinear combination of two input vectors. We note that, while bilinear operators are not a new development (e.g., they have been widely used in a variety of systems engineering and control applications for many years [30]), the current paper shows how to chain these bilinear components together into a powerful new predictive model.

Let $q_t$ and $o_t$ be the state and observation at time $t$. Let $W$ be a 3-mode tensor, and let $q$ be a vector. The 1-layer state update for a PSRNN is defined as:

$$q_{t+1} = \frac{W \times_2 o_t \times_3 q_t + b}{\|W \times_2 o_t \times_3 q_t + b\|_2} \tag{5}$$

Here the 3-mode tensor of weights $W$ and the bias vector $b$ are the model parameters. This architecture is illustrated in Figure 1a. It is similar, but not identical, to the PSR update (Eq. 1); sec 3.1 gives

more detail on the relationship. This model may appear simple, but crucially the tensor contraction $W \times_2 o_t \times_3 q_t$ integrates information from $b_t$ and $o_t$ multiplicatively, and acts as a gating mechanism, as discussed in more detail in section 5.

The typical approach used to increase modeling capability for BFs (including PSRs) is to use an initial fixed nonlinearity to map inputs up into a higher-dimensional space [31, 30]. PSRNNs incorporate such a step, via RFFs. However, a multilayered architecture typically offers higher representation power for a given number of parameters [32].

To obtain a multilayer PSRNN, we stack the 1-layer blocks of Eq. (5) by providing the output of one layer as the observation for the next layer. (The state input for each layer remains the same.) In this way we can obtain arbitrarily deep RNNs. This architecture is displayed in Figure 1b.

We choose to chain on the observation (as opposed to on the state) as this architecture leads to a natural extension of 2SR to multilayered models (see Sec. 4.2). In addition, this architecture is consistent with the typical approach for constructing multilayered LSTMs/GRUs [12]. Finally, this architecture is suggested by the full normalized form of an HSE PSR, where the observation is passed through two layers.

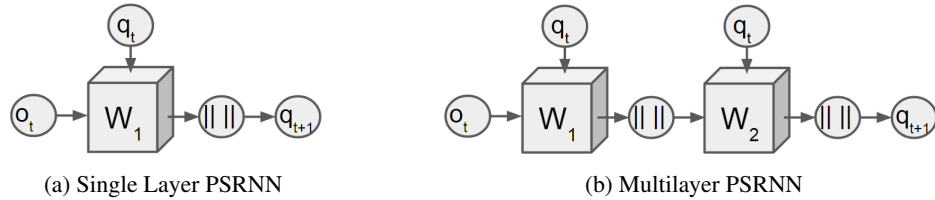

(a) Single Layer PSRNN              (b) Multilayer PSRNN

Figure 1: PSRNN architecture: See equation 5 for details. We omit bias terms to avoid clutter.

## 4.2 Learning PSRNNs

There are two components to learning PSRNNs: an initialization procedure followed by gradient-based refinement. We first show how a statistically consistent 2SR algorithm derived for PSRs can be used to initialize the PSRNN model; this model can then be refined via BPTT. We omit the BPTT equations as they are similar to existing literature, and can be easily obtained via automatic differentiation in a neural network library such as PyTorch or TensorFlow.

The Kernel Bayes Rule portion of the PSR update (equation 1) can be separated into two terms: $(W \times_3 q_t)$ and $(Z \times_3 q_t)^{-1}$. The first term corresponds to calculating the joint distribution, while the second term corresponds to normalizing the joint to obtain the conditional distribution. In the discrete case, this is equivalent to dividing the joint distribution of $f_{t+1}$ and $o_t$ by the marginal of $o_t$; see [33] for details.

If we remove the normalization term, and replace it with two-norm normalization, the PSR update becomes $q_{t+1} = \frac{W \times_3 q_t \times_2 o_t}{\|W \times_3 q_t \times_2 o_t\|}$, which corresponds to calculating the joint distribution (up to a scale factor), and has the same functional form as our single-layer PSRNN update equation (up to bias).

It is not immediately clear that this modification is reasonable. We show in appendix B that our algorithm is consistent in the discrete (realizable) setting; however, to our current knowledge we lose the consistency guarantees of the 2SR algorithm in the full continuous setting. Despite this we determined experimentally that replacing full normalization with two-norm normalization appears to have a minimal effect on model performance prior to refinement, and results in improved performance after refinement. Finally, we note that working with the (normalized) joint distribution in place of the conditional distribution is a commonly made simplification in the systems literature, and has been shown to work well in practice [34].

The adaptation of the two-stage regression algorithm of Hefny et al. [7] described above allows us to initialize 1-layer PSRNNs; we now extend this approach to multilayered PSRNNs. Suppose we have learned a 1-layer PSRNN $P$ using two-stage regression. We can use $P$ to perform filtering on a dataset to generate a sequence of estimated states $\hat{q}_1, ..., \hat{q}_n$. According to the architecture described in Figure 1b, these states are treated as observations in the second layer. Therefore we can initialize the second layer by an additional iteration of two-stage regression *using our estimated*

*states $\hat{q}_1, ..., \hat{q}_n$ in place of observations.* This process can be repeated as many times as desired to initialize an arbitrarily deep PSRNN. If the first layer were learned perfectly, the second layer would be superfluous; however, in practice, we observe that the second layer is able to learn to improve on the first layer's performance.

Once we have obtained a PSRNN using the 2SR approach described above, we can use BPTT to refine the PSRNN. We note that we choose to use 2-norm divisive normalization because it is not practical to perform BPTT through the matrix inverse required in PSRs: the inverse operation is ill-conditioned in the neighborhood of any singular matrix. We observe that 2SR provides us with an initialization which converges to a good local optimum.

## 4.3 Factorized PSRNNs

In this section we show how the PSRNN model can be factorized to reduce the number of parameters prior to applying BPTT.

Let $(W, b_0)$ be a PSRNN block. Suppose we decompose $W$ using CP decomposition to obtain

$$W = \sum_{i=1}^{n} a_i \otimes b_i \otimes c_i$$

Let $A$ (similarly $B$, $C$) be the matrix whose $i$th row is $a_i$ (respectively $b_i$, $c_i$). Then the PSRNN state update (equation (5)) becomes (up to normalization):

$$q_{t+1} = W \times_2 o_t \times_3 q_t + b \tag{6}$$
$$= (A \otimes B \otimes C) \times_2 o_t \times_3 q_t + b \tag{7}$$
$$= A^T (Bo_t \odot Cq_t) + b \tag{8}$$

where $\odot$ is the Hadamard product. We call a PSRNN of this form a *factorized PSRNN*. This model architecture is illustrated in Figure 2. Using a factorized PSRNN provides us with complete control over the size of our model via the rank of the factorization. Importantly, it decouples the number of model parameters from the number of states, allowing us to set these two hyperparameters independently.

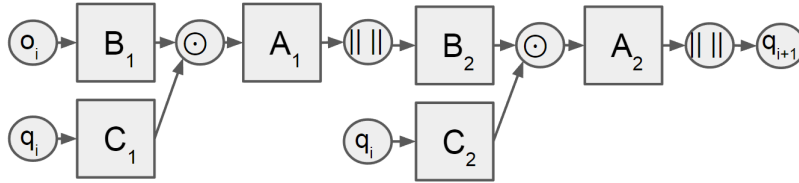

Figure 2: Factorized PSRNN Architecture

We determined experimentally that factorized PSRNNs are poorly conditioned when compared with PSRNNs, due to very large and very small numbers often occurring in the CP decomposition. To alleviate this issue, we need to initialize the bias $b$ in a factorized PSRNN to be a small multiple of the mean state. This acts to stabilize the model, regularizing gradients and preventing us from moving away from the good local optimum provided by 2SR.

We note that a similar stabilization happens automatically in randomly initialized RNNs: after the first few iterations the gradient updates cause the biases to become non-zero, stabilizing the model and resulting in subsequent gradient descent updates being reasonable. Initialization of the biases is only a concern for us because we do not want the original model to move away from our carefully prepared initialization due to extreme gradients during the first few steps of gradient descent.

In summary, we can learn factorized PSRNNs by first using 2SR to initialize a PSRNN, then using CP decomposition to factorize the tensor model parameters to obtain a factorized PSRNN, then applying BPTT to the refine the factorized PSRNN.

# 5 Discussion

The value of bilinear units in RNNs was the focus of recent work by Wu et al [35]. They introduced the concept of Multiplicative Integration (MI) units — components of the form $Ax \odot By$ — and showed that replacing additive units by multiplicative ones in a range of architectures leads to significantly improved performance. As Eq. (8) shows, factorizing $W$ leads precisely to an architecture with MI units.

Modern RNN architectures such as LSTMs and GRUs are known to outperform traditional RNN architectures on many problems [12]. While the success of these methods is not fully understood, much of it is attributed to the fact that these architectures possess a gating mechanism which allows them both to remember information for a long time, and also to forget it quickly. Crucially, we note that PSRNNs also allow for a gating mechanism. To see this consider a single entry in the factorized PSRNN update (omitting normalization).

$$[q_{t+1}]_i = \sum_j A_{ji} \left( \sum_k B_{jk}[o_t]_k \odot \sum_l C_{jl}[q_t]_l \right) + b \tag{9}$$

The current state $q_t$ will only contribute to the new state if the function $\sum_k B_{jk}[o_t]_k$ of $o_t$ is non-zero. Otherwise $o_t$ will cause the model to forget this information: the bilinear component of the PSRNN architecture naturally achieves gating.

We note that similar bilinear forms occur as components of many successful models. For example, consider the (one layer) GRU update equation:

$$z_t = \sigma(W_z o_t + U_z q_t + c_z)$$
$$r_t = \sigma(W_r o_t + U_r q_t + c_r)$$
$$q_{t+1} = z_t \odot q_t + (1 - z_t) \odot \sigma(W_h o_t + U_h(r_t \odot q_t) + c_h)$$

The GRU update is a convex combination of the existing state $q_t$ and and update term $W_h o_t + U_h(r_t \odot q_t) + c_h$. We see that the core part of this update term $U_h(r_t \odot q_t) + c_h$ bears a striking similarity to our factorized PSRNN update. The PSRNN update is simpler, though, since it omits the nonlinearity $\sigma(\cdot)$, and hence is able to combine pairs of linear updates inside and outside $\sigma(\cdot)$ into a single matrix.

Finally, we would like to highlight the fact that, as discussed in section 5, the bilinear form shared in some form by these models (including PSRNNs) resembles the first component of the Kernel Bayes Rule update function. This observation suggests that bilinear components are a natural structure to use when constructing RNNs, and may help explain the success of the above methods over alternative approaches. This hypothesis is supported by the fact that there are no activation functions (other than divisive normalization) present in our PSRNN architecture, yet it still manages to achieve strong performance.

# 6 Experimental Setup

In this section we describe the datasets, models, model initializations, model hyperparameters, and evaluation metrics used in our experiments.

We use the following datasets in our experiments:

- **Penn Tree Bank (PTB)** This is a standard benchmark in the NLP community [36]. Due to hardware limitations we use a train/test split of 120780/124774 characters.

- **Swimmer** We consider the 3-link simulated swimmer robot from the open-source package OpenAI gym.[3] The observation model returns the angular position of the nose as well as the angles of the two joints. We collect 25 trajectories from a robot that is trained to swim forward (via the cross entropy with a linear policy), with a train/test split of 20/5.

- **Mocap** This is a Human Motion Capture dataset consisting of 48 skeletal tracks from three human subjects collected while they were walking. The tracks have 300 timesteps each, and are from a Vicon motion capture system. We use a train/test split of 40/8. Features consist of the 3D positions of the skeletal parts (e.g., upper back, thorax, clavicle).

- **Handwriting** This is a digit database available on the UCI repository [37, 38] created using a pressure sensitive tablet and a cordless stylus. Features are $x$ and $y$ tablet coordinates and pressure levels of the pen at a sampling rate of 100 milliseconds. We use 25 trajectories with a train/test split of 20/5.

Models compared are LSTMs [30], GRUs [13], basic RNNs [14], KFs [3], PSRNNs, and factorized PSRNNs. All models except KFs consist of a linear encoder, a recurrent module, and a linear decoder. The encoder maps observations to a compressed representation; in the context of text data it can be viewed as a word embedding. The recurrent module maps a state and an observation to a new state and an output. The decoder maps an output to a predicted observation.[4] We initialize the LSTMs and RNNs with random weights and zero biases according to the Xavier initialization scheme [39]. We initialize the the KF using the 2SR algorithm described in [7]. We initialize PSRNNs and factorized PSRNNs as described in section 3.1.

In two-stage regression we use a ridge parameter of $10^{(-2)}n$ where $n$ is the number of training examples (this is consistent with the values suggested in [8]). (Experiments show that our approach works well for a wide variety of hyperparameter values.) We use a horizon of 1 in the PTB experiments, and a horizon of 10 in all continuous experiments. We use 2000 RFFs from a Gaussian kernel, selected according to the method of [28], and with the kernel width selected as the median pairwise distance. We use 20 hidden states, and a fixed learning rate of 1 in all experiments. We use a BPTT horizon of 35 in the PTB experiments, and an infinite BPTT horizon in all other experiments. All models are single layer unless stated otherwise.

We optimize models on the PTB using Bits Per Character (BPC) and evaluate them using both BPC and one-step prediction accuracy (OSPA). We optimize and evaluate all continuous experiments using the Mean Squared Error (MSE).

## 7 Results

In Figure 3a we compare performance of LSTMs, GRUs, and Factorized PSRNNs on PTB, where all models have the same number of states and approximately the same number of parameters. To achieve this we use a factorized PSRNN of rank 60. We see that the factorized PSRNN significantly outperforms LSTMs and GRUs on both metrics. In Figure 3b we compare the performance of 1- and 2-layer PSRNNs on PTB. We see that adding an additional layer significantly improves performance.

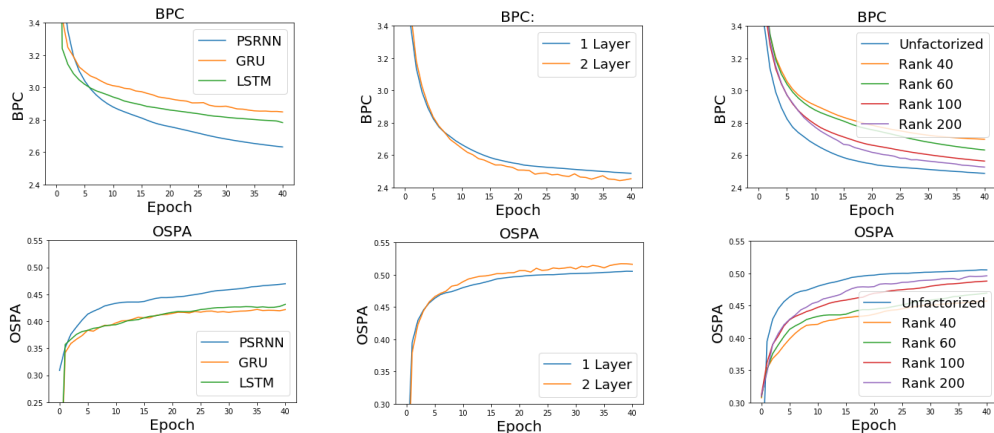

(a) BPC and OSPA on PTB. All models have the same number of states and approximately the same number of parameters.

(b) Comparison between 1- and 2-layer PSRNNs on PTB.

(c) Cross-entropy and prediction accuracy on Penn Tree Bank for PSRNNs and factorized PSRNNs of various rank.

Figure 3: PTB Experiments

In Figure 3c we compare PSRNNs with factorized PSRNNs on the PTB. We see that PSRNNs outperform factorized PSRNNs regardless of rank, even when the factorized PSRNN has significantly more model parameters. (In this experiment, factorized PSRNNs of rank 7 or greater have more model parameters than a plain PSRNN.) This observation makes sense, as the PSRNN provides a simpler optimization surface: the tensor multiplication in each layer of a PSRNN is linear with respect to the model parameters, while the tensor multiplication in each layer of a Factorized PSRNN is bilinear. In addition, we see that higher-rank factorized models outperform lower-rank ones. However, it is worth noting that even models with low rank still perform well, as demonstrated by our rank 40 model still outperforming GRUs and LSTMs, despite having fewer parameters.

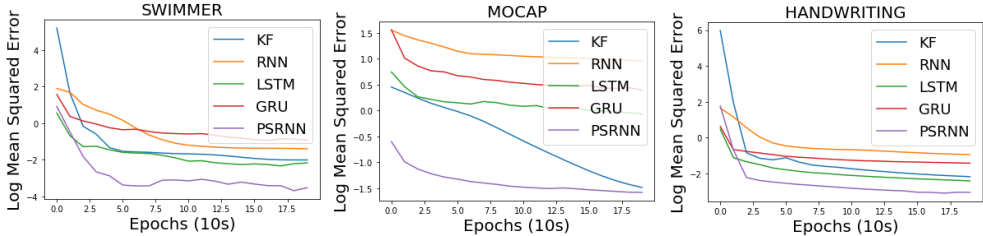

(a) MSE vs Epoch on the Swimmer, Mocap, and Handwriting datasets

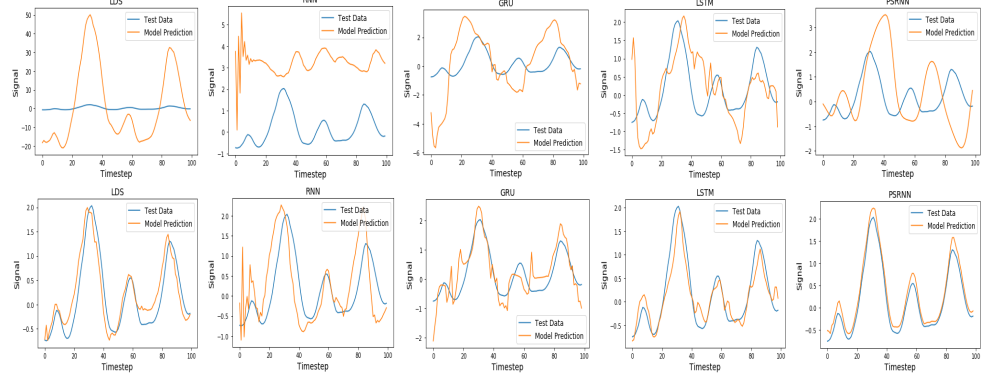

(b) Test Data vs Model Prediction on a single feature of Swimmer. The first row shows initial performance. The second row shows performance after training. In order the columns show KF, RNN, GRU, LSTM, and PSRNN.

Figure 4: Swimmer, Mocap, and Handwriting Experiments

In Figure 4a we compare model performance on the Swimmer, Mocap, and Handwriting datasets. We see that PSRNNs significantly outperform alternative approaches on all datasets. In Figure 4b we attempt to gain insight into why using 2SR to initialize our models is so beneficial. We visualize the the one step model predictions before and after BPTT. We see that the behavior of the initialization has a large impact on the behavior of the refined model. For example the initial (incorrect) oscillatory behavior of the RNN in the second column is preserved even after gradient descent.

# 8   Conclusions

We present PSRNNs: a new approach for modelling time-series data that hybridizes Bayes filters with RNNs. PSRNNs have both a principled initialization procedure and a rich functional form. The basic PSRNN block consists of a 3-mode tensor, corresponding to bilinear combination of the state and observation, followed by divisive normalization. These blocks can be arranged in layers to increase the expressive power of the model. We showed that tensor CP decomposition can be used to obtain factorized PSRNNs, which allow flexibly selecting the number of states and model parameters. We showed how factorized PSRNNs can be viewed as both an instance of Kernel Bayes Rule and a gated architecture, and discussed links to existing multiplicative architectures such as LSTMs. We applied PSRNNs to 4 datasets and showed that we outperform alternative approaches in all cases.

**Acknowledgements**   The authors gratefully acknowledge support from ONR (grant number N000141512365) and DARPA (grant number FA87501720152).

## Footnotes

[1]For convenience we assume that the system is $k$-observable: that is, the distribution of all future observations is determined by the distribution of the next $k$ observations. (Note: not by the next $k$ observations themselves.) At the cost of additional notation, this restriction could easily be lifted.

[2]Note that we can train a regression model to predict any quantity from the state. This is useful for general sequence-to-sequence mapping models. However, in this work we focus on predicting future observations.

[3] https://gym.openai.com/

[4]This is a standard RNN architecture; e.g., a PyTorch implementation of this architecture for text prediction can be found at `https://github.com/pytorch/examples/tree/master/word_language_model`.

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
