[Supplementary Material]

# A  Learning HSE-PSRs with Two-Stage Regression

In this section we derive the initialization algorithm described in section 3.2. We follow the two-stage regression formulation described in [7]. We denote by $\omega_t$, $\eta_t$, and $\phi_t$ the feature representation of the observation at time $t$, a window of past observations ending at time $t-1$ and a window of future observations starting at $t$. In this work, the feature representation is obtained by applying random Fourier features[28] for an RBF kernel followed by linear projection through PCA.

## A.1  Hilbert Space Embedding of Distributions

We will briefly describe the concept of Hilbert space embedding of distributions, which is the machinery that we use to represent distributions and perform Bayesian state updates. We refer the reader to [26] for more details on this topic.

Let $\phi(.)$ be the feature map of a kernel $\kappa$ such that $\kappa(x_1, x_2) = \langle \phi(x_1), \phi(x_2) \rangle$. For a random variable $X$ with a distribution $\mathbb{P}(X)$, the corresponding *mean map* is defined as

$$\mu_X = E_{x \sim \mathbb{P}}[\phi(x)]$$

For a universal kernel such as RBF kernel on a compact set, it can be shown that $\mathbb{P}(X)$ uniquely determines the mean map $\mu_X$. In other words, $\mu_{\mathcal{X}}$ is a sufficient representation of the distribution $\mathbb{P}(X)$.

Another important quantity is the covariance operator $\mathcal{C}_{YX} \equiv E_{x,y \sim \mathbb{P}(X,Y)}[\phi_X(x) \otimes \phi_Y(y)]$. The covariance operator is a sufficient representation of the joint distribution $\mathbb{P}(X, Y)$. Note that we can use different kernels with different feature maps for $X$ and $Y$.

The covariance operator allows us to perform conditioning. Given that $X = x$ we can compute the mean map for the distribution $\mathbb{P}(Y|X = x)$ using kernel Bayes rule:

$$\mu_{Y|x} = \mathcal{C}_{YX} \mathcal{C}_{XX}^{-1} \phi_X(x) \tag{10}$$

It is beneficial to consider these concepts in the special case of discrete variables and delta kernel. In this case, the feature function $\phi$ is the indicator function, the mean map is a probability vector and the covariance operator is a matrix encoding a joint probability table.

## A.2  Random Fourier Features

Random Fourier features is a method for obtaining an approximate finite-dimensional feature map $\hat{\phi}$ such that $\hat{\phi}(x_1)^\top \hat{\phi}(x_2) \approx \kappa(x_1, x_2)$. For the Gaussian RBF kernel this function is given by:

$$\hat{\phi}(x) \equiv \sqrt{\frac{2}{D}} [\cos(x^\top v^{(1)}), \sin(x^\top v^{(1)}), \ldots \tag{11}$$

$$\cos(x^\top v^{(D)}), \sin(x^\top v^{(D)})]^\top \tag{12}$$

where $v^{(i)}$ are i.i.d Gaussian variables. In order to reduce dimensionality, we use the feature map $\tilde{\phi}(x) = U^\top \hat{\phi}(x)$ where $U^\top$ is a projection matrix obtained via PCA.

## A.3  State Representation and Updates

In this we consider $k$-observable systems, where it is sufficient to maintain the distribution of future $k$ observations in order to make future predictions without the need to look back into history. We write $\phi_t \equiv \phi(o_{t:t+k-1})$ to indicate the application of future feature function on a window of $k$ observations starting at $t$. We define the predictive state $q_t = \mathbb{E}[\phi_t \mid o_{1:t-1}]$. As discussed in the previous section, this is a sufficient representation of the distribution of future observations.

An HSE-PSR is parameterized by two linear operators represented by tensors $W$ and $Z$ such that

$$\mathcal{C}_{o_t \phi_{t+1} | o_{1:t-1}} = W \times_3 q_t \tag{13}$$

$$\mathcal{C}_{o_t \phi_{t+1} | o_{1:t-1}} = Z \times_3 q_t \tag{14}$$

By plugging in (14) into kernel Bayes rule (10) we easily get the state update equation (1).

### A.4 Two Stage Regression

We now derive the algorithm for initializing $W$. Initializing $Z$ is similar. Define $\zeta_t = \text{vec}(\phi_{t+1} \otimes \omega_t)$, where $\text{vec}$ denotes reshaping into a vector. With an sbuse of notation we can write

$$E[\zeta_t \mid o_{1:t-1}] = WE[\phi_t \mid o_{1:t-1}]$$

Taking the expectation of both sides w.r.t $\eta_t$ gives

$$E[\zeta_t \mid \eta_t] = WE[\phi_t \mid \eta_t]$$
$$\mathcal{C}_{\zeta\eta}\mathcal{C}_{\eta\eta}^{-1}\eta_t = W\mathcal{C}_{\phi\eta}\mathcal{C}_{\eta\eta}^{-1}\eta_t \quad \forall \eta_t$$
$$\mathcal{C}_{\zeta\eta} = W\mathcal{C}_{\phi\eta}$$
$$\mathcal{C}_{\zeta\eta}\mathcal{C}_{\phi\eta}^{+} = W$$

Equation (3) is the simply the result of replacing the above covariances with their empirical estimates and using a tensor notation instead of vectorizing outer products.

## B   On the Consistency of Initialization

In this section we provide a theoretical justification for the PSRNN. Specifically, we show that in the case of discrete observations and a single layer the PSRNN provides a good approximation to a consistent model. We first show that in the discrete setting using a matrix inverse is equivalent to a sum normalization. We subsequently show that, under certain conditions, two-norm normalization has the same effect as sum-normalization. .

Let $q_t$ be the PSR state at time $t$, and $o_t$ be the observation at time $t$ (as an indicator vector). In this setting the covariance matrix $C_t = E[o_t \times o_t | o_{1:t-1}]$ will be diagonal. By assumption, the normalization term $Z$ in PSRs is defined as a linear function from $q_t$ to $C_t$, and when we learn PSRN by 2-stage regression we estimate this linear function consistently. Hence, for all $q_t$, $Z \times_3 q_t$ is a diagonal matrix, and $(Z \times_3 q_t)^{-1}$ is also a diagonal matrix. Furthermore, since $o_t$ is an indicator vector, $(Z \times_3 q_t)^{-1} \times_2 o_t = o_t/P(o_t)$ in the limit. We also know that as a probability distribution, $q_t$ should sum to one. This is equivalent to dividing the unnormalized update $\hat{q}_{t+1}$ by its sum. i.e.

$$q_{t+1} = \hat{q}_{t+1}/P(o_t)$$
$$= \hat{q}_{t+1}/(\mathbf{1}^{\top}\hat{q}_{t+1})$$

Now consider the difference between the sum normalization $\hat{q}_{t+1}/(\mathbf{1}^{\top}\hat{q}_{t+1})$ and the two-norm normalization $\hat{q}_{t+1}/\|\hat{q}_{t+1}\|_2$. Since $q_t$ is a probability distribution, all elements will be positive, hence the sum norm is equivalent to the 1-norm. In both settings, normalization is equivalent to projection onto a norm ball. Now let $S$ be the set of all valid states. Then if the diameter of $S$ is small compared to the distance from (the convex hull of) $S$ to the origin then the local curvature of the 2-norm ball will be negligible, and both cases will be approximately equivalent to projection onto a plane. We note we can obtain an $S$ with this property by augmenting our state with a set of constant features.

## C   A Summary of Learning PSRNNs

Here we provide a concise summary of the training steps for PSRNNs. A Python implementation is available at `https://github.com/cmdowney/psrnn`.

1. Collect training data as triplets $(h_t, o_t, o_{t:t+k-1}, o_{t+1:t+k})$ (i.e. history, observation, future and shifted future).

2. Determine observation kernel bandwidth $s$ using median trick.

3. Sample i.i.d Gaussian vectors $v^{(i)}$ for $i = 1, 2, \dots, D$ with standard deviation $1/s$. Use these vectors to computer RFF feature map for observations using (12). Use PCA to obtain a lower-dimensional feature map $\omega$.

4. Repeat the previous step on futures and histories to obtain feature maps $\phi$ and $\eta$.

5. Use (3) to initialize the parameter tensor $W$.

6. Use backpropagation through time through the update equation $q_{t+1} = \frac{W \times_3 q_t \times_2 o_t}{\|W \times_3 q_t \times_2 o_t\|}$ to refine $W$.