[Reviews · NeurIPS 2017]

Reviewer 1



This paper proposes a new model for dynamical systems (called PSRNN), which combines the frameworks of PSR and RNN non-trivially. The model is learned from data in two steps: The first step initialize the model parameters using two-stage regression (2SR), a method previously proposed by Hefny et al for learning PSRs. The second step use Back-propagation-through-time to refine the parameters. The learned model can then be used for filtering and prediction. The model has an appealing bi-linear gating mechanism, resembling the non-linear gating mechanisms used in LSTM and other models and enjoys rich functional form via kernel embedding and/or multilayer stacking. A factorized version is also discussed. The empirical study in this paper shows a clear advantage of PSRNN over previously proposed methods and looks quite promising. Unfortunately, the paper is not well written, not self-contained and hard to follow. There is no chance I could implement this algorithm given the description in the paper. I often find this very frustrating in practice. Most prominently, the introduction of 2SR in Section 4.1 is highly unclear. First, I could not make mathematical sense out of equations (4) and (5). How does a multiplication of 3-mode tensor and a matrix is defined? Going over the referred paper of Hefny et al (briefly) I could not readily determine how their approach can be adapted to PSRNN in this paper. This should be discussed in detail. In particular, I didn't understand how W,Z in (4) and (5) are used to produce a PSRNN, since Z is not used in a PSRNN model. So why one need to learn Z? In line 109, the definition of f_{t+k} is ambiguous. How is it precisely defined in terms of o_{1:T}? Although Section 6 is named "Theoretical discussion", no theory on the performance of PSRNN is given in this paper. The last paragraph of this section is unclear to me. To conclude, the model and results in this paper should be definitely published, but after a major revision of the manuscript, in particular making the learning procedure clearer to the non-expert reader.

Reviewer 2



This paper extends predictive state representations (PSRs) to a multilayer RNN architecture. A training algorithm for this model works in two stages: first the model is initialized by two-stage regression, where the output of each layer serves as the input for the next layer, next the model is refined by back propagation through time. The number of parameters can be controlled by factorizing the model. Finally, the performance of this model is compared to LSTMs and GRUs on four datasets. The proposed extension is non-trivial and seem to be useful, as demonstrated by the experimental results. However, the initialization algorithm, which is a crucial learning stage, should be described more clearly. Specifically, the relation of the estimation equations (3-5) to Heffny et al. is not straightforward and requires a few intermediate explanations. On the other hand, Section 3 can be shortened. The introduction of the normalization in Equation (1) is a bit confusing as it takes a slightly different form later on. Two strong aspects of the paper are that initialization algorithm exploits a powerful estimation procedure and the Hilbert embedding provides a rich representation. More generally, exploiting existing approaches to learn dynamical systems is an interesting avenue of research. For example, [Haarjona etal. 2016] proposed construction of RNNs by combining neural networks with Kalman Filter computation graphs. In the experimental section, can the authors specify what kernels have been used? It would be interesting to elaborate on the added value of additional layers. The experiments show the added value of two-layer models with respect to a one-layer in terms of test performance measure. Is this added value achieved immediately after initialization, or only after applying BPTT? Does the second layer improve by making the features richer? Does the second stage regression learn the residuals of the first one. Minor comments and typos: - Figures are too small - Line 206: delete one ‘the’ Ref: Haarjona etal. 2016, Backprop KF: Learning Discriminative Deterministic State Estimators

Reviewer 3



The authors study the well known Predictive State Representation problem and introduce a new neural network based architecture for tackling this problem. They discuss pros and cons of Bayesian approaches while mentioning the power and drawbacks of RNN based setting. Later, they propose a new procedure which tries to exploit the power of both settings. The paper is fairly readable and managed in a good way while covering many pieces. The used techniques are theoretically well studied in the literature and the authors provide a good intuition for their proposal. There are 3 minor things I like to see in the paper. It would be interesting to have the computation complexity of the initialization. It would be good to cite this paper "Tensor Regression Networks" https://arxiv.org/pdf/1707.08308.pdf which uses the tensor contraction as a core of deep learning process. You use "can be " two times in the abstract.